# Social tipping dynamics in the energy system

Floor Alkemade[1], Bart de Bruin[1], Amira El-Feiaz[1], Francesco Pasimeni[1], Leila Niamir[2], Robert Wade[1]

[1]Eindhoven University of Technology, Eindhoven, The Netherlands
[2]IIASA, Vienna, Austria

*Correspondence to*: Floor Alkemade (f.alkemade@tue.nl)

**Abstract.** This paper reviews evidence on how the fast growth in renewables can trigger social tipping dynamics which potentially might accelerate a system-wide energy transition. It does so by reviewing a variety of literature across several disciplines addressing socio-technical dimensions of energy transitions. The tipping dynamics in wind and solar power create potential for cascading effects to energy demand sectors, including household energy demand. These most likely start with
10 shift actions and adoption of household-scale batteries and heat pumps. Key enablers are strong regulations incentivising reductions in demand and setting minimum efficiency levels for buildings and appliances. While there is evidence of spillovers to more environmentally friendly behaviour, the extent of these and the key leverage points present a knowledge gap. Moreover, these behavioural feedback loops require strong additional policy support to 'make them stick'. Understanding the economic and social tipping dynamics in a system can empower decision-makers, fostering realistic energy transition policies.
The paper highlights energy communities as a promising niche for leveraging tipping dynamics. Ultimately, bridging the gap between these tipping dynamics and institutional reforms is crucial for unlocking the full potential of sustainable energy systems.

## 1 Introduction

A transition from a fossil-fuel-based energy system to an energy system based on renewable energy sources is key to meeting climate targets. This energy transition involves interdependent changes to technologies and infrastructures, to the behaviour of firms and individuals, and to institutions and governance. That is, energy transitions are socio-technical transitions (Geels et al., 2017). Historical case studies, for example, of the transition from wood to coal, argue that energy transitions typically
take decades and have severe disruptive effects, affecting the livelihood of many people (Freeman & Louçã, 2002). Both the fear of these negative consequences and the lock-in of the current fossil-fuel-based system are given as explanations for the slow pace of current-day sustainability transitions (Hughes, 1993; Negro et al., 2012).

This view of energy transitions as inevitably slow processes has recently been challenged. First, we now have some examples of relatively fast energy transitions, e.g., to natural gas in The Netherlands or to combined heat and power in Denmark
(Sovacool, 2016). Second, the diffusion of renewable energy technologies like wind and solar has been much faster than anticipated by energy transition scenarios (Creutzig et al., 2017; de Coninck et al., 2018.; Trutnevyte et al., 2019; Wilson et al., 2013).

Social tipping dynamics, in analogy to the tipping dynamics of ecological systems, have received increased attention as a possible mechanism that potentially explains this acceleration of the transition to more sustainable socio-technical systems
(Otto et al., 2020). Social tipping dynamics for sustainability occur in social-environmental systems with alternative stable states, where a change process unfolds rapidly (or nonlinearly), driven by feedback mechanisms and with some degree of irreversibility or stickiness (Milkoreit, 2022; Global Tipping Report 2023). Several social factors can initiate social tipping dynamics, including tipping in costs and prices, norms and behaviour and policy (Roberts et al. 2018, Otto et al. 2020). When

these dynamics drive the system in a more desirable direction, such as a more sustainable state, these dynamics are labelled "positive" social tipping dynamics. Compared to, for instance, ecological tipping dynamics, positive social tipping is thus frequently framed as normatively desirable and intentionally activated or triggered (Lenton et al, 2022; Milkoreit, 2022).

Systems Dynamics modelling can capture the effects of feedback and interactions on system behaviour. Here, feedbacks are generally understood as circular causal processes where the effect of change in one part of a system leads to further change in that part. When an increase in $x$ leads to further increases in $x$ (or when a decrease leads to further decrease) through this circular chain of causality, this is known as positive or reinforcing feedback. Negative or balancing feedbacks occur when an increase in $x$ leads to a decrease in $x$ (or vice versa) and negative feedbacks are therefore associated with stability (Meadows, 2008). Furthermore, when feedback-powered tipping dynamics spread from one system to another one or upwards to drive system change at a higher scale, tipping cascades can occur (Sharpe & Lenton, 2021). Therefore, while feedback loops are not as fundamental as other leverage points for sustainability, which focus on system goals and paradigms (Meadows, 2008), the relatively minor efforts triggering tipping dynamics nonetheless hold the potential to trigger deeper system change through cascading interactions.

In the energy system, the cost reduction in renewable energy technologies is a driver for social tipping dynamics. As solar and wind energy sources become prevalent in the energy system, their costs decrease, enabling wider adoption (Söderholm & Klaassen 2007, Way et al 2022). This, in turn, leads to economies of scale, further reducing costs and creating positive feedback loops that drive even more installations (Isoard & Soria 2001). In economic terms, tipping occurs when the cost of renewable energy becomes competitive with or even lower than that of conventional energy sources. The solar energy sector in Germany presents a prominent example of positive social tipping: When strong public, policy and industry support aligned simultaneously with a strong decrease in support for nuclear energy, this led to unexpected and fast price performance improvements and demand increases in solar technology, boosting the sector globally. These reinforcing feedbacks are weakened by balancing feedbacks that dampen the growth of renewables, and that prevent system change. These balancing feedbacks can originate from vested interests in the fossil-fuel-based system but also from barriers encountered by renewables.

Tipping dynamics are observed within various subsystems of energy systems (Geels & Ayoub 2023). These dynamics can occur when radical and incremental technological innovations move the system towards cleaner and more efficient energy production and consumption. Such dynamics can act as catalysts for rapid changes and start cascading effects within the energy landscape. But tipping dynamics can also occur within the realm of actors and institutions, where changes in policies, regulations, market dynamics, or in the choices and behaviours of firms and individuals can have large effects on the trajectory of the energy system (Otto et al. 2020). The importance of such social and behavioural factors, like policy support, societal acceptance or changing norms, is extensively reported in descriptive case studies that are the foundation of the field of sustainability transitions research (Köhler et al., 2019). For the energy system, the challenge is to connect the current tipping dynamics in low-level intervention points to higher-level intervention points to realise tipping cascades that fundamentally change the system.

This review paper, therefore, addresses the following question: How can the fast growth in renewables start system-wide tipping cascades that accelerate the energy transition? To this end, the paper reviews the sustainability transitions literature for feedback and interactions in the socio-technical energy system that may build on the tipping dynamics in the supply of renewable energy to create tipping cascades that in turn lead to a transition of the energy system. We make use of causal loop diagrams to visualise reinforcing (R) and balancing (B) feedback processes.

Section 2 first discusses how the fast growth in renewable electricity supply may initiate further tipping processes, in technologies, household energy behaviours, and throughout the socio-technical system. Here, specific attention is given to the electrification of households, to avoid-shift-improve (ASI) measures for demand reduction and to how sustainable lifestyles, and the social and political system can generate tipping dynamics in the energy system. Section 3 then explores energy communities as an area where positive tipping dynamics hold great potential. Finally, section 4 concludes.

## 2 Fast growth in renewable electricity supply drives social tipping in the energy system

Most evidence on tipping dynamics in energy systems concerns the price performance of new technologies (Otto et al. 2020). Renewables are now among the cheapest energy generation options. Cost reductions in renewable generation technologies like wind energy and solar photovoltaics (PV) have been massive and much faster than predicted. The price of electricity from solar energy declined by 89% from 2009 to 2019, and the price of wind energy declined by 70% in this period. In some contexts, cost-parity in energy generation for wind and solar has been reached or even exceeded, making them cheaper than

fossil generation (Haegel et al. 2019, IRENA 2022a,b). For wind and solar energy generation, the main reinforcing feedback (denoted by R in the causal loop diagrams in Figure 1) that created these tipping dynamics is cost reduction and performance improvement through economies of learning and economies of scale, leading to more deployment and, in turn, to more learning (Sharpe & Lenton 2021; Kavlak et al., 2018; Nemet & Greene, 2022).

       The diffusion of solar PV is also analysed as a social process where considerations of observability and trialability and processes
like word-of-mouth play a role next to costs and performance (Rogers 2003, Bollinger & Gillingham 2012, Palm 2017, Rode & Weber 2016). Adoption of rooftop solar PV, for instance, is typically clustered in space, where people are more likely to adopt when people nearby also have adopted (Graziano & Gillingham 2015, van der Kam et al. 2018). Therefore, more adoption leads to increased observability and trialability (i.e., learning), which in turn leads to more adoption. Moreover, markets are still expanding as performance improvements make the technology attractive to a wider range of users. As a result of these

technological improvements and cost reductions, renewable generation is increasingly possible in locations where wind or sun conditions are less favourable or where installation is more difficult and costly, as demonstrated by the increasing attention for floating solar (Gonzalez-Sanchez et al. 2021, Jin et al. 2023).

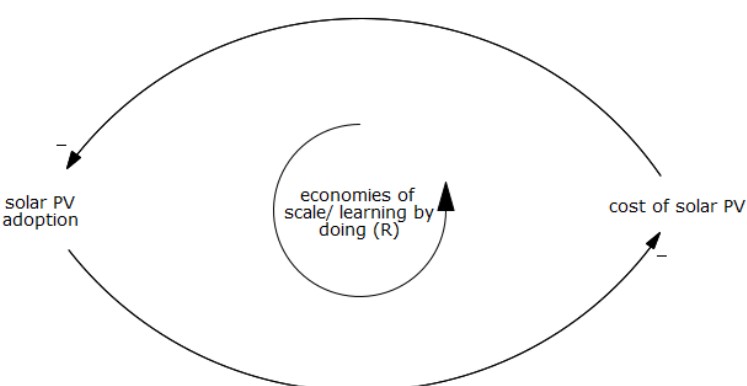

**Figure 1: Economies of scale and learning in solar PV.** As more solar PV is adopted, costs are reduced to due economies of scale and learning effects, in turn driving up further solar PV adoption

### 2.1. Household electrification.

       The goal of energy systems is to provide energy services to end users. The main energy uses are for heat and electricity in industry and buildings and for transport. The industrial, residential and transport sectors together account for 70 per cent of the
total global electricity consumption in 2019, and these sectors also are responsible for approximately 60 per cent of the worldwide carbon dioxide ($CO_2$) emissions (IEA, 2021a; IEA, 2023a). The decarbonisation of the energy system is thus a key driver of overall decarbonisation efforts.

       In end-use sectors, the tipping dynamics in wind and solar may initiate further decarbonisation of the energy system through electrification of energy demand. These developments are typically supported by government subsidies and regulation on the

energy performance of buildings, and households that electrify their demand may do so for both environmental and financial reasons. Moreover, the fast cost reductions as observed in wind and solar are more likely to occur in smaller and modular technologies (Wilson et al. 2020). In consumer end-use sector, there are several other small and modular technologies that may reach cost-parity in the short term, like electric vehicles, household batteries and heat pumps (Meldrum et al. 2023).

The transportation sector is a relevant example of these advancements. The increasing prevalence of electric cars, along with other electricity-powered alternatives such as e-bikes, e-scooters, and other mopeds, indicates the key role of batteries in novel modular demand and the significant contribution to sector-wide decarbonisation. In addition to facilitating emission-free mobility, the batteries in electric vehicles can also support the grid infrastructure during periods of ample electricity generation from renewable sources by functioning as modular storage systems.

Stationary household batteries are specifically attractive in places where feed-in tariffs for solar energy into the grid are much lower than the retail price for energy from the grid. The large-scale adoption of household batteries may influence the decarbonisation of the energy system in two ways. First, it reduces curtailment of household solar PV generation, better matching renewable energy supply with demand. Second, it reduces grid congestion during peaks in solar generation (reinforcing feedback on the left in Figure 2). Increasingly, grid congestion is a barrier to further grid integration of renewables. However, few countries currently have strong incentives in place to stimulate demand to synchronise with the availability of renewable energy supply.

Another area of electrification of the residential sector are heating and cooling systems (Figure 2). Heat demand is often met by natural gas boilers. Based on IEA (2022) analysis, natural gas accounts for 42% of global heating energy demand, with a 40% share of the heating mix in the European Union and over 60% in the United States. When low-carbon heat sources like waste heat are available, this can be a preferred option. When this is not the case, electrification of heating demand through heat pumps can lead to a large reduction in energy demand. This shift to low-carbon heat sources requires additional changes in technologies and infrastructure in houses and neighbourhoods. The availability of low-cost and abundant renewable energy supply is a key enabler here, and early adopters are often households that generate excess electricity with their own rooftop PV systems. Another important enabler is increased insulation (also to reduce overall heat demand). For instance, the adoption of solar PV coupled with efficiency and sufficiency measures may both yield increased adoption of heat pumps, as shown at the top in Figure 2. Barriers to the electrification of heating are the lack of technologies for heat storage and the cumbersome installation process. A more radical and politically challenging behavioural change would be to provide incentives to live in smaller homes or to have higher occupancy per dwelling, for example in planning decisions.

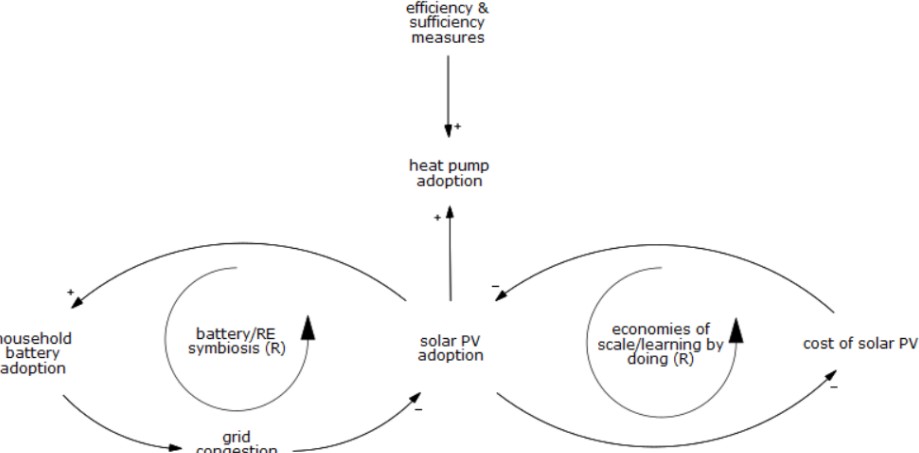

**Figure 2: Feedback loop in electrification of heating & cooling.** The reinforcing feedback between cost reduction and increased adoption of solar PV may trigger another reinforcing feedback of adoption of household battery and reduction in

grid congestion, resulting into further solar PV adoption. Furthermore, solar PV adoption supports demand for heat pumps, boosted by adequate efficiency and sufficiency measures

## 2.2. Household energy demand reduction

While for many regions in the world renewable energy potential exceeds demand, a fast energy transition faces constraints regarding the availability and sustainable sourcing of materials and personnel (Wang et al., 2023). Most scenarios, therefore, envision a reduction of demand where the demand for energy should be brought in line with what can be sustainably produced in the short term. Indeed, reducing energy demand is key in 1.5°C pathways (Koide et al., 2021), especially in wealthy countries. At the same time, energy access and service provision will need to grow for many less-developed countries, and for
poor people everywhere to ensure decent living standards and wellbeing (IPCC, 2022a). Although we observe a decoupling of energy demand and income in some places, in general, household energy demand grows with income. Pro-environmental attitudes and behaviour have also been correlated with income, further complicating the challenge of how to reduce income inequality and material and energy consumption to sustainable sufficiency levels (Du et al., 2022). Moreover, individuals with high socio-economic status (top 10 per cent) are responsible for a large share of emissions (IPCC 2022b; IEA, 2021b). These
individuals could have a large positive impact when they reduce GHG emissions, becoming role models of low-carbon lifestyles, investing in low-carbon businesses, and advocating for stringent climate policies (Creutzig et al. 2022). Such approaches are also discussed in the context of energy justice and equitable energy demand reduction (Büchs et al., 2023).

However, demand reduction options are often constrained by the existing socio-technical system. It is, for example, difficult
for individuals to change their mobility practices, when employer preference regarding workplace presence do not change. The Avoid-Shift-Improve (ASI) framework (Creutzig et al. 2022) is often used to identify demand reduction options. Avoid options reduce unnecessary energy consumption, possibly by redesigning service provisioning systems. Shift refers to the switch to already existing competitive, efficient and cleaner technologies and service provisioning systems. Improve refers to efficiency improvements in existing technologies. While improve options are not sufficient to tip the energy system to a
decarbonised state, they are an important enabler for options that can. Figure 3 adds ASI measures as an additional element in previous figure on feedback loop in electrification of heating & cooling.

The different ASI measures often co-occur. While avoid options have the largest mitigation potential, they often need to be flanked with shift and improve options to be attractive. For example, when people switch from natural gas heating to heat pumps, good insulation (improve) is a condition. Typically avoid and shift options require larger changes in social practices
and in the broader socio-technical system compared to improve options. More specifically, options where both behavioural and technological change is required, or that require a substantial change in social and user practices, are typically more difficult to realise and thus difficult as a starting point for tipping dynamics (Geels et al. 2018).

Higher prices lead to reduced energy demand, providing evidence for measures like a carbon tax. Natural gas consumption in the EU and in the period August 2022 to January 2023 decreased by 19% compared to the average gas consumption for the
180 same months in the previous 5 years. However, this also came with increased levels of energy poverty, particularly affecting low-income households in badly insulated homes (IEA 2023). Interestingly the high prices also triggered and opened the opportunity for sufficiency-based energy price interventions. Because of the relationship between income and energy use, a rebound effect may occur (see top right balancing feedback in Figure 3) when technologically or socially induced demand reductions lead to a higher budget and more energy demand (Newell et al. 2021; van den Bergh 2011; Sorrell et al. 2020).

At the individual and household level, energy behaviour changes, including turning down the thermostat and reducing the demand for hot tap water (shorter showers), are effective strategies (Roy et al. 2012, Creutzig et al. 2016, Ivanova et al. 2020). These strategies are most effective when combined with policy support and shift and improve measures, in many countries regulation regarding the energy performance of buildings and subsidies for energy saving measures are enablers of such behaviour change. More specifically, digital technologies are key to better matching renewable supply with demand to avoid
curtailments and grid congestion (load shifting and balancing) but have not yet reached widespread diffusion. Digitalisation in general (at the top-left in Figure 3) can also play a key role in avoiding unnecessary energy demand (Wilson et al. 2020).

Empirical studies show that informing people about the energy conservation behaviours of their neighbours combined with the public labelling of energy conservation behaviour as desirable, can lead to significant reductions in energy consumption behaviour (Gockeritz et al. 2010, Allcott 2011, Horne & Kennedy 2017, Bonan et al. 2020). These studies show that a relatively weak form of sanctioning (i.e., showing approval and disapproval of particular behaviour by using thumps up/down or positive and negative smileys), already has a modest positive effect on energy savings. Peer effects in social network structures can provide inhibiting or supporting conditions for the diffusion of energy conservation practices, depending on the structure of the network and the type of activity (Wolske et al. 2020).

Changes in the energy behaviour and lifestyle of individuals can make a large contribution to sustainability but are only feasible when supported by changes in the broader socio-technical system (Nisa et al. 2019, Niamir et al. 2020). This means that social tipping of energy consumption by individuals, households or organisations is conditioned by a range of factors such as social and cultural norms, ownership and control of resources, technology accessibility, infrastructure design and services availability, social network structures, and organisational resources, knowledge and awareness and sociodemographic characteristic (Ameli and Brandt, 2015; Niamir et al., 2020; Steg et al. 2018). Social and behavioural change is thus constrained by the existing socio-technical system and people's daily lives and behaviour, or social practices (Matthews & Wynes 2022). Social practices approaches shine a light on the culturally embedded routines which reproduce (but also potentially transform) socio-technical energy systems from the bottom up. Crucially, they also point to the differentiation of these practices across social groups (e.g., women versus men, upper class versus working class) (Husu, 2022).

Avoid options reduce unnecessary energy consumption. But, when avoiding energy use is undesirable from a well-being perspective, then shifting the way this activity is done (or finding an alternative means to the same goal) is key. When the demand reductions stem from changes in norms or behaviours with a sustainability motive, the risks of rebound effects are lower. However, different attitudes make some demand-side alternatives difficult to scale up in the population (Geels 2023). Often lacking are enabling conditions for just and smooth change, as for instance city infrastructure or the built environment may prevent people from avoiding using private cars instead of alternatives like walking, cycling, or taking public transport.

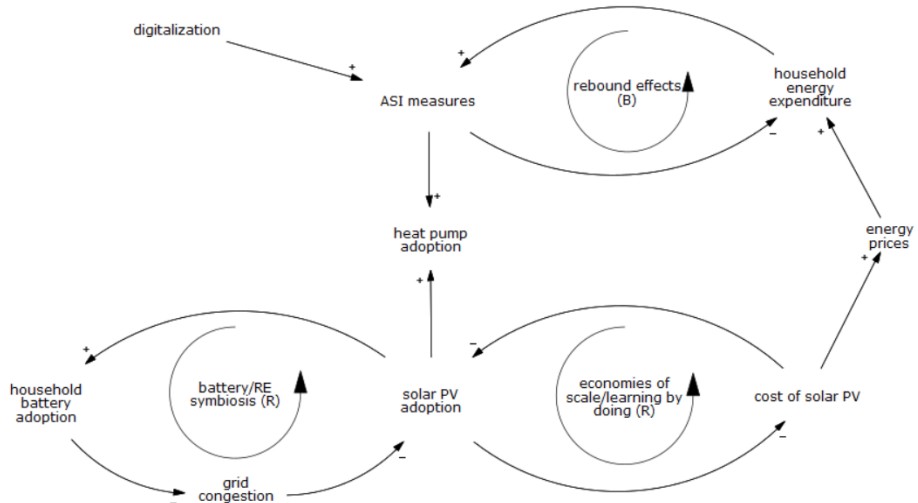

Figure 3: Feedback processes in reducing energy demand

While some demand-side behaviour changes are quite swift, a key policy challenge is how to make the new and desired behaviour 'stick'. An example is the substantial energy demand reduction in Europe in the winter of  2022/2023, resulting from concerns about high energy prices and the war in Ukraine (IEA, 2023). Similarly, but at a global system level, in 2020, the world witnessed a reduction in global fossil fuel emissions as a result of COVID-19 lockdowns across the globe. However, emissions rebounded in 2021, reaching levels comparable to those observed in 2019 (LeQuere et al. 2021, Friedlingstein et al. 2022). These observations reinforce that social tipping dynamics are tipping dynamics rather than tipping points (Milkoreit et al. 2018, Geels & Ayoub 2023), not just because they take some time to evolve but also because different reinforcing processes are needed to provide momentum (Hughes, 1987) and to ensure that the change sticks or becomes embedded or irreversible on the relevant time scales.

## 2.3. Sustainable lifestyles, and social and political tipping dynamics

Interestingly, pro-environmental behaviours may induce other pro-environmental behaviours, so changes in behaviour in mobility, or food may spill over to energy behaviours (Steg & Vlek 2009, Steg 2023). The adoption of household PV for environmental reasons may thus induce other pro-environmental behaviours. When the new sustainable lifestyle and behaviour becomes common and the norm starts to shift, this also increases the political feasibility of strict regulation. There is, for example, public support for measures like incentives towards renewable technology and a ban on least energy-efficient household appliances (see Figure 4).

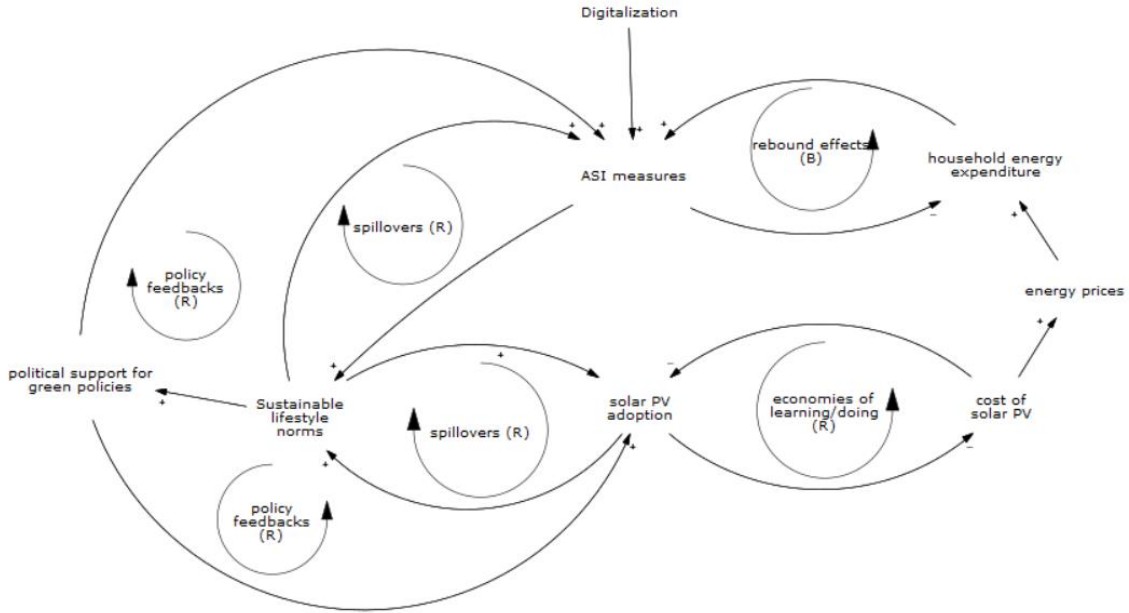

**Figure 4. Social and political dynamics in energy system tipping**

The positive feedback loop mechanism of opinion exchange can increase awareness and promote more sustainable lifestyles. However, it can also have a negative effect when contrarians get the majority in a given social group, leading to the amplification and reinforcement of anti-environmental beliefs. For this reason, avoiding opinion polarisation is crucial in climate-related issues to foster cohesion for effective government action (Badullovich 2023, Mayer & Smith 2023). Citizens'

environmental consciousness and the formation of their opinions directly affects actions that impact the local and global
environment (Chung et al. 2019, van den Bergh et al. 2019).

The presence of a group with strong anti-environmental beliefs can discourage pro-environmental engagement and support for climate change initiatives. Opinion polarisation makes it challenging to reach consensus and decreases public support for environmental initiatives, posing a challenge for policymakers (Maertens et al. 2020). To mitigate negative feedback loop and harness the positive cascade effect of opinion dynamic, some governments have implemented policies to incentivize pro-
environmental behaviours, while awareness campaigns and education aim to correct misinformation and provide accurate information (Charlier & Kirakozian 2020, Baiardi 2022). When opinions drive clique formation, they can lead to concrete pro-environment actions, such as social movements and support for climate change initiatives (Winkelmann et al. 2022).

Social acceptance and changes in norms and behaviours, may have large influence on both direct consumer demand and policy support (Edelenbosch et al., 2018; Nemet, 2006). Civil society engages with energy transitions in many ways: from adopting
energy efficient technology, to joining energy cooperatives; from environmental activism to resistance against wind parks (Chilvers et al., 2021). These interactions are driven by (changes in) perceptions, attitudes, motivations, emotions, beliefs, values, and norms (Clayton et al., 2015), sometimes triggered by external events like the oil crisis or nuclear accidents. Some of these factors also may influence the willingness to adopt a certain technology (as in Edelenbosch et al., 2018), adoption or societal acceptance is not only driven by price.
There is extensive literature on the social acceptance of renewable energy infrastructure (Batel, 2020; Ellis & Ferraro, 2017; Wolsink, 2018). One of the most prominent conceptualisations of social acceptance is Wüstenhagen et al.'s (2007) social acceptance triangle, comprising community, market and socio-political acceptance. This draws attention to the fact that community acceptance or local opposition to projects can influence general public or political acceptance, and societal demand
for renewable energy. From this perspective, demand is not simply the economic behaviour of individuals or households but is a product of societal relations. One potential balancing feedback for renewables deployment is project delays caused by local opposition, which leads to pressure to streamline planning and reduce participation options, which in turn creates more opposition. This dynamic is seen in many EU countries today.

Finally, policy feedbacks are well recognised in political science literature, whereby policy not only stimulates deployment but also creates legitimacy and new interests, leading to increased lobbying and support for policy to support the new industries and further deployment (Hess 2016, Meckling et al. 2017, Meckling 2019, Roberts et al. 2018, Rosenbloom et al. 2019, Sewerin et al. 2020, Fesenfeld et al. 2022). For example, Kelsey (2021) identifies 'green spirals' which resembled tipping dynamics for the reduced use of CFCs for ozone protection. Policies engendered new industrial interests who in turn support new policies.
Kelsey also identified that these spirals can transcend domestic politics and scale up to the international level. This is similar to the notion of tipping cascades.

Key considerations for policymakers hoping to create tipping dynamics in this way is the sequencing of policies (Meckling et al, 2017). For the energy transition, similar dynamics can potentially be found with the renewables industry. For instance, the
German feed-in tariff for renewables is frequently mentioned as an enabling condition for this feedback (Otto et al. 2020, Nijsse et al., 2023). Further, strong pro-environment policies may incentivise firms towards more and R&D and innovation, thereby expanding industrial sectors for low-carbon technologies. In this way, public opinion may also increase support and acceptance for new low-carbon technologies, increasing pressure on policymakers in setting-up goals and strategies for a more sustainable society (Geels and Ayoub, 2023).

The political sphere can not only trigger social tipping but can also tip itself into a new state, generating a tipping cascade (Stadelmann-Steffen et al. 2021, Eder & Stadelmann-Steffen 2023). Indeed, the same applies to any group, organisation or institution that is part of the socio-technical system. For example, civil society could also be a key element in energy system tipping dynamics. Increasing attention is being paid to prosumerism which can be understood as a broad movement towards a decentralized democratic energy model (Campos & Marín-González, 2020). These and other civil society movements interact
with the state, which in turn creates opportunities or barriers to different lines of action for citizens or households, engendering balancing or reinforcing policy feedbacks.

While research on policy feedbacks frequently targets its findings towards policymakers, this knowledge can also be used by civil society or interest coalitions to try to initiate such feedback processes. Indeed, some research from social movements theory identifies movement-policy feedbacks or 'opportunity/threat' spirals in which "demands lead to concessions that encourage further demands, and so on" (Biggs, 2003, p. 228; McAdam et al., 2001). Winkelmann et al. (2022) discuss the relationship between the Fridays for Future movement and European states in ways which could align with this idea. Focussing specifically on energy, such feedbacks could help to explain the recent boom of the energy cooperative movement in countries like the Netherlands, for example.

## 2.4 Balancing feedbacks

In general, the energy transition requires a system-level transformation of the energy system which depends on both phasing out fossil fuels, and accelerating renewable energy provision. In the previous sections, we highlighted the promise of positive social tipping dynamics in renewables development while recognising that this is inevitably an incomplete picture without fully considering the fossil phase out. Below we address some of the feedbacks (Figure 5) that strengthen the existing energy system, (see Eker & Wilson (2022) for a systematic overview).

First, sources of balancing feedback, lock-in, and path dependence of fossil fuel-based energy systems are energy infrastructures, technologies and institutions (Hughes 1987; Dangerman & Schellnhuber 2013; Kohler et al. 2019). These can directly hinder the decarbonisation of the energy system through existing standards and resistance from incumbents and vested interests. Further, renewable energy generation sometimes faces curtailment, and the mismatch of renewable supply with energy demand slows down the replacement of fossil fuels. Indirectly, the availability of cheap energy has stimulated demand for energy-intensive goods and services. Similarly, the high return on fossil fuel investments and the assessment of renewables as risky make it difficult to move capital from fossil to renewables (Pauw et al. 2022). As an example, in the early 2000s, the UK government provided initial capital grants to boost offshore wind demonstration projects. This has, in turn, built confidence among financial investors, easing access to resources for project developers (i.e., lower interest rates) (Kern et al. 2014; Geels & Ayoub 2023), stimulating the overall offshore sector.

Second, examples of barriers encountered by renewables are challenges related to intermittency and the need for a flexible and well-managed grid infrastructure to ensure a reliable and stable energy supply. The increasing need to electrify various end-user sectors (IRENA 2023) adds further complexity to the grid management challenge. For instance, the electrification of transportation is experiencing rapid growth, boosted by policy initiatives for the adoption of e-mobility. Similarly, there is a strong policy focus on electrifying heating and cooling systems in residential areas and districts. Moreover, the electrification of demand is not always viable, and the energy transition may negatively impact individuals with restricted financial resources (Sovacool et al. 2019). In addition, many processes that reinforce fossil-fuel-based energy systems, ranging from subsidies to vested interests and existing infrastructures, are still in place. Energy infrastructures are typically built for a lifespan of around 40 years, and changing these infrastructures takes place on the timescale of months to years. Once built, they contribute to stabilising the system state and are a source of path dependence and lock-in.

Social dynamics can also create balancing feedbacks when they mobilise opposition and a lack of societal support for larger-scale solar and wind parks (Devine-Wright 2007; Klok et al. 2023; Kluskens et al. forthcoming; Windemer 2023). Therefore, cost-competitiveness is not a sufficient indicator to predict support for technologies for which the main public concerns are about spatial impacts, health and safety, and questions of fairness. This shows that economic tipping dynamics alone are not sufficient to realise rapid decarbonisation.

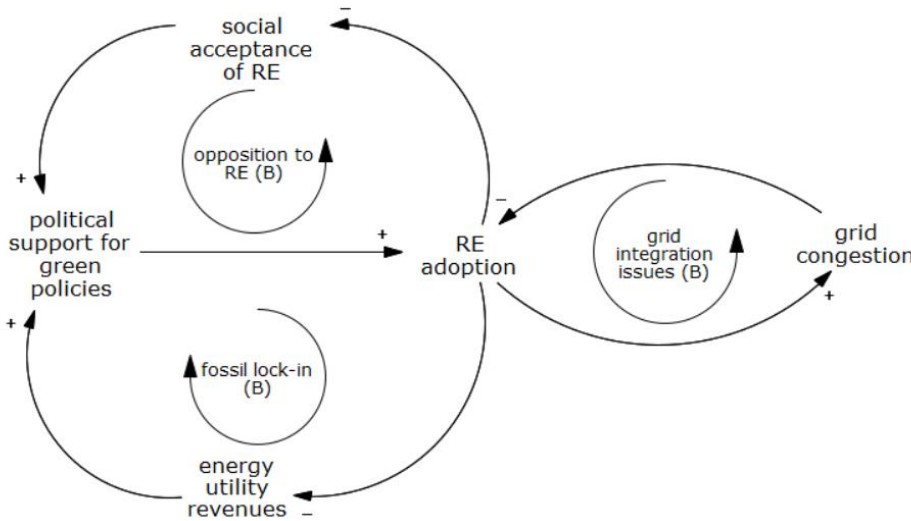

**Fig. 5. Balancing feedbacks for renewable energy adoption.**

## 3. Tipping dynamics in Energy Communities

While there is thus potential for tipping dynamics in technology adoption, the balancing feedbacks regarding system integration and social practices hamper the scale-up to tipping cascades. Or in system dynamics terms, the dynamics remain restricted to low-level leverage points or feedback loops (Meadows, 2008). This section explores energy communities as a social innovation which targets higher level leverage points such as system rules and goals at both micro/meso (e.g., community) and macro levels (e.g., policy supports). They do so by changing the institutional environment in which individuals or other actors operate, 340 which can lead to a strengthening or weakening of balancing and reinforcing feedbacks described above. They can also lead to the creation or removal of certain feedbacks under new system conditions. Energy communities can thus strengthen, weaken, add or remove the feedbacks discussed in previous sections.

Many energy communities take the form of renewable energy cooperatives. A renewable energy cooperative is a bottom-up, legally registered collective of citizens that aims to create social, environmental and/or economic benefits for its members 345 through energy-related activities (Doci et al., 2015; van Summeren et al., 2020; Hicks & Ison, 2018). Many cooperatives are local enterprises with diversified activity portfolios (Reis et al., 2021). They create value for their members via energy-related projects, ranging from awareness raising to cooperative energy production (Oteman et al., 2014). In the EU, the Clean Energy Package, adopted in 2019, aims for a central role for these cooperatives in decarbonising the energy system. More specifically, it advocates energy cooperatives as a way to enable citizens to participate in and benefit from the energy transition. Renewable 350 energy cooperatives have increased in scale, scope and number throughout European member states (Blasch et al., 2021; REScoop.eu, 2020).

Energy communities often have social and sustainability goals as a main objective, for example to reduce dependence on the centralised energy infrastructure, while also taking advantage of the possibility to produce, consume and sell the energy produced back to the grid (Yildiz et al. 2015, Bauwens et al. 2016, Bauwens et al. 2022). Other objectives include to reduce 355 energy poverty and to accelerate decarbonisation of the energy system via the spread of renewable energy solutions (Shapira et al. 2021). Typical characteristics of energy communities are voluntary and open membership (van den Berghe and Wieczorek, 2022), the 'one member – one vote' principle (Wierling et al., 2023), a high degree of community ownership and governance, and fair value distribution (Mourik et al., 2020). Activities of renewable energy cooperatives include collective

energy generation and selling, collective purchasing of renewable energy, consulting and awareness raising (Gui & MacGill, 2018) and development & ownership of energy projects (Wierling et al., 2023). In addition, some cooperatives also offer (peer-to-peer) trading of energy balancing and flexibility services (van Summeren et al., 2020; Verkade & Höffken, 2019).

People join an energy community for a variety of reasons including self-interests but also because of social cohesion and sense of community (Albinsson & Perera 2012). In order to maintain long-term stability, strong motivation is often required by key project leaders. Shared social norms, values, trust, and collaboration among members also contribute to this attempt (Schoor & Scholtens 2015). This often creates challenges when communities grow in size (Barnes et al. 2022). By increasing in size, an energy community becomes too large to be smoothly organised and managed, leading also to business models that deviate from the original idea of polycentricity and equity (Blasch et al. 2021, Anfinson et al. 2023).

Energy communities' cooperative and legal structures often require that any profits are re re-invested in the community, further stimulating investment in clean energy technologies. The electrification of residential districts can then also create a positive feedback loop into the adoption of home storage systems and other sustainable choices. Especially communities that strive for energy autonomy or independence from the grid reduce grid congestion, even if they do not actively offer flexibility to the grid (see reinforcing feedback loop on the right in Figure 5). Secondly, energy communities are found to be more accepted and supported by local citizens (Hogan et al., 2022; Jobert et al., 2007; Musall et al., 2011; Rogers et al., 2008; Strachan et al., 2015; Warren and McFadyen, 2010), which can in turn influence broader socio-political acceptance. While energy communities might face equal pushback from incumbent utilities, this increased community and socio-political acceptance might also buffer against this.

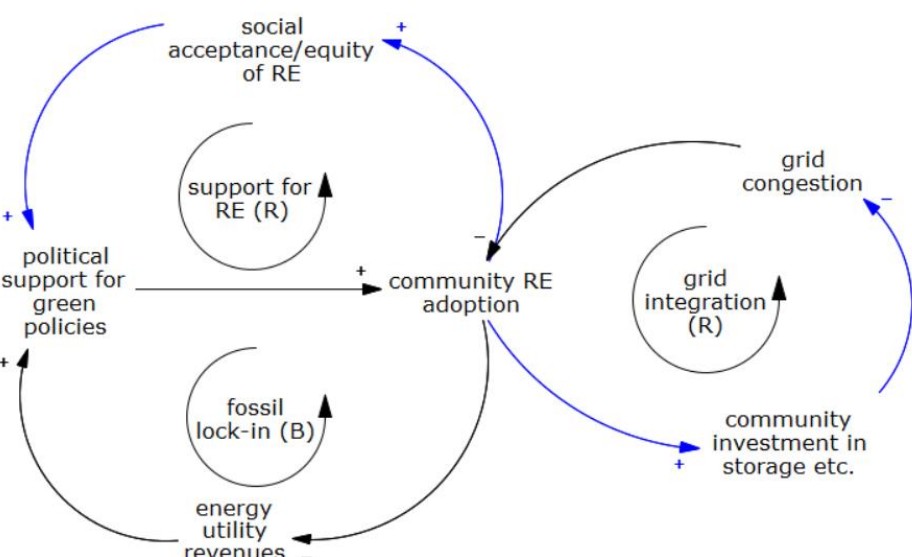

**Fig. 6. Energy communities can turn balancing feedbacks into reinforcing feedbacks for RE adoption.** Blue arrows indicate new mechanisms arising from the institutional context of renewable energy.

Embracing community values and norms can also function as an external incentive for behaviour change and can increase the adoption rate of sustainable practices (Smith. et al. 2020, Manfredo et al., 2017). The rise of community energy within western Europe is an example of embedding sustainable behaviour within the existing motivation mechanisms of individuals. Where within the former fossil-fuel-based centralised energy systems were aimed at pursuing energy security (i.e., achieving

affordable, available, acceptable and accessible energy for all members of society Cherp & Jewell (2014)), the technological innovation of affordable small-scale technologies could suddenly fulfil the existing desires and demands for democracy, autarky, justice and social cohesion (Brown et al. 2020, van de Poel & Taebi 2022). Once new behaviour is adopted, the engagement in such energy community practices can lead to a positive feedback loop between sustainable behaviour (Sloot et al. 2018) and the prioritisation of ecosystem system conservation-related values (Radke et al. 2022).

Together with socio-environmental motivations, the economic component is among the main factors increasing the willingness to participate in an energy community (Heuninckx et al. 2022). For instance, a home storage system may not be affordable some households and sharing practices in energy communities enhance affordability and access to essential goods and services (Watson 2004). The demand for privately owned goods leads to inefficient consumption and excessive production (Baudrillard 2016, Frenken & Schor 2017), contradicting the United Nations' Sustainable Development Goal number 12, which emphasises doing more with fewer resources. Instead, participation into an energy community can help transitioning from individual to shared ownership and consumption of goods, thereby enabling sustainable consumption while also increasing empowerment, reciprocity and energy democracy (Pasimeni 2021, Dudka et al. 2023, Ivanova &Buchs 2023). Moreover, studies have demonstrated that shared ownership decreases the demand for individually owned goods, creating a positive feedback loop where changes in demand (but not reduction) prompt corresponding adjustments in the supply side (Pasimeni & Ciarli 2023). For instance, when participation in an energy community motivates people to share also (electric) vehicles this will result in using fewer cars, reducing production and the overall environmental impact (Nematchoua et al. 2021, Belmar et al. 2023).

To summarise, energy communities are in line with sustainable goals and targets, while also addressing economic considerations for households facing financial constraints. Moreover, as energy communities have the potential to expand into providing other sustainable goods and services, they align with the sufficiency logic (Thomas et al. 2019) and polycentric systems of governance (Ostrom 2010). These communities, especially those aiming for complete autonomy from centralised energy systems, operate differently from traditional market-based organisations. Communities operate outside the dynamics driven solely by price concerns and instead prioritise energy independence, social cohesion, and community well-being (Hasanov & Zuidema 2018). This approach may lead to more sustainable lifestyles and an overall reduction in fossil fuel consumption, although it remains uncertain whether energy communities will also result in a decrease in overall energy consumption.

## 4. Discussion and Conclusions

The tipping dynamics observed in the wind and solar power sectors have the potential to trigger cascading effects throughout energy demand sectors, including household energy consumption. This transformative process is likely to start with shift actions, such as the adoption of household scale batteries and heat pumps, thereby enhancing less energy-intensive lifestyles. These actions will modify energy demand and improve energy service efficiency, which are instrumental in accelerating the decarbonisation of our energy system. Nevertheless, a strong regulatory framework is crucial to the speed of this transition as it can incentivise reductions in energy demand and set minimum efficiency standards for buildings and appliances. By doing so, regulation becomes a key enabler of positive tipping points in the adoption of novel technologies and behaviours, facilitating the shift to more sustainable practices.

Although spillover effects are observed, as adoption of environmentally friendly behaviours seems to increase, a substantial knowledge gap exists. Specifically, it is important to understand the extent of these spillovers and the key leverage points. Research efforts must be dedicated to shedding light on the connections between individual actions and systemic change. Moreover, behavioural feedback loops, once identified, require policy support to "make them stick". Strengthening the connection between individual choices and institutional reforms requires effort to bridge effectively these two levels of influence.

In this complex landscape, energy communities emerge as an attractive and rapidly growing niche. Communities are likely to boost widespread adoption of renewable energy technologies, and have fundamentally different goals and operating principles compared to incumbent actors locked-in the centralised energy system. Energy communities are therefore high-impact leverage points, capable of catalysing significant changes in the energy landscape.

By looking deeper into the dynamics of renewable energy adoption and behavioural shifts, it becomes clear that bridging the gap between tipping dynamics and institutional reforms is pivotal to unlocking the full potential of sustainable energy systems. This can be addressed at several scales. For example, the relationship between community energy and behavioural tipping dynamics and spillovers is one potential area of future investigation. Furthermore, we can also ask how community energy as a social innovation can cascade upwards and tip higher level or coupled systems.

**Competing interests** The contact author has declared that none of the authors has any competing interests

**Acknowledgements** This work was supported by the European Union (ERC, FAST, 101044076).

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
