# Peer review of "Social tipping dynamics in the energy system"

_Earth System Dynamics, 2023_

## Author Response (AR1)

Dear editors, dear reviewers,

On behalf of the authoring team, I would like to thank the reviewers for their constructive feedback. Both reviewers provided general suggestions regarding the overall line of argumentation and more detailed comments. We have implemented the revisions suggested below, this has led to a quite substantial rewrite of the whole paper.

Regarding the general feedback, the paper is currently submitted as a review paper. In the revision we have now described (in introduction) that we review the sustainability transitions literature. We have also explained how we use system dynamics approaches and causal loop diagrams to describe and visualize these feedbacks including a short explanation of the terminology.

Regarding the suggestion of the first reviewer, the revision has tried to link the section on energy communities more clearly to the overall argumentation of the paper.

In addition to these main points, we have addressed the following other points.

Detailed comments:

1) Abstract – we will update the abstract to better fit the revised methods and provide additional detail.
2) Regarding the definition of social tipping points – we aim to use harmonized definitions throughout the papers in this special section. But we will elaborate a bit on the choices in definition and will especially follow up the reviewer's suggestion to explicitly address any differences with the use of tipping point in climate science. This indeed especially pertains to the potential irreversibility (less final in social processes) but also to the timescales.
3) Following the suggestion of the reviewer we would add the following references in the revised paper:

   Söderholm, P., Klaassen, G. Wind Power in Europe: A Simultaneous Innovation–Diffusion Model. *Environ Resource Econ* 36, 163–190 (2007).

   Way, R., Ives, M. C., Mealy, P., & Farmer, J. D. (2022). Empirically grounded technology forecasts and the energy transition. *Joule*, *6*(9), 2057-2082.

4) In the revised version we would add a methods section that more clearly explains our approach to this review and our conceptual use of system dynamics to structure the outcomes of the review. This will include a short description of the main terms including references.
5) Thank you for pointing this out. We will carefully revise the paper to address this.
6) Thank you for pointing this out. In the revision we will explicitly address the difference between tipping points and tipping dynamics.
7) This sentence is indeed better placed elsewhere. We will include the discussion on leverage points in the general part of the paper where we describe how we use system dynamics concepts to study tipping dynamics.
8) Similarly to point 7, we will include this point in the main conceptual and methods part and will explicitly refer back in some of the examples and the section on energy communities.
9) We propose to rephrase to "In some contexts, cost-parity in energy generation for wind and solar has been reached or even exceeded, making them cheaper than fossil generation."
10) This is indeed confusing, we propose to revise to "The increasing attention for floating solar (Gonzalez-Sanchez et al. 2021, Jin et al. 2023)"
11) We propose to revise to "The political sphere can also be seen as a tipping element itself, as it not only can trigger social tipping but can also tip itself into a new state, generating a tipping cascade (Stadelmann-Steffen et al. 2021, Eder & Stadelmann-Steffen 2023). Indeed, the same applies to any group, organisation or institution that is part of the socio-technical system. For example, civil society could also be a key element in energy system tipping dynamics."
12) We will review the sentence clearly distinguishing between electrification of heating and other sustainable heating solutions.

13) We will remove the mention of solar home systems in the global south and revise to connect smaller elements more clearly to the overall argumentation.
14) Thank you for pointing this out, we will carefully revise the figures, captions and their referencing in the text
15) Thank you for pointing this out, we will carefully revise the figures, captions and their referencing in the text
16) We will elaborate a bit on this point in the revision
17) Thank you for pointing this out, we will carefully revise the figures, captions and their referencing in the text
18) This is indeed inconsistent, we propose to remove 'isolated'
19) We agree and will revise accordingly
20) Here we will elaborate a bit more on the financial and non-financial motives for people to join energy communities and add references
21) We will include discussion in the conclusion section and also add a paragraph on leverage points.

Dear reviewer, dear editor,

On behalf of the authoring team, I would like to thank the reviewers for their constructive feedback. Both reviewers provide general suggestions regarding the overall line of argumentation and more detailed comments.

Regarding the general feedback, the paper is currently submitted as a review paper. In a revision, we would make our approach to this method more clear. In short, we review the sustainability transitions literature for interactions that may contribute to feedback loops, or for feedback loops. We also aim to present empirical evidence for individual interactions. In some cases, this empirical material comes from case studies in the sustainability transition literature, but in some cases also from more controlled experiments in environmental psychology. We indeed use a conceptual system dynamics approach to organise and evaluate these feedback loops. In the revision, we will make this approach more explicit, distinguishing more clearly between interacting processes and feedback loops.

Regarding the suggestion of the first reviewer, the revision will seek to strengthen the argumentation towards energy communities, although, given the review character of this paper, we also refer to future work on this topic.

In addition to these main points, we provide a more detailed response to the numbered reviewer comments below.

Detailed comments:

1) As the focus of the paper is indeed on the reinforcing feedback loops in renewables supply and not on demand reduction in fossil, we will follow the suggestions of the reviewer to rewrite the abstract.
2) In the revised version we would add a methods section that more clearly explains our approach to this review and our conceptual use of system dynamics to structure the outcomes of the review. This will include a short description of the main terms including references.
3) Thank you for pointing this out, we will carefully revise the figures, captions and their referencing in the text
4) Thank you for this suggestion which we will adopt in the revision
5) Thank you for pointing this out, we will carefully revise the figures, captions and their referencing in the text
6) Thank you for this suggestion, we will include it in the revised version and follow up on the suggested references